# OpenReview forum: "Nested Gloss Makes Large Language Models Lost"
_ICLR.cc/2025/Conference — ICLR 2025 Conference Withdrawn Submission_

### Official Review · Reviewer_RKh4 · 2024-10-20

**Soundness:** 3
**Presentation:** 3
**Contribution:** 2
**Rating:** 5
**Confidence:** 4

**Summary:**

This paper describes DeepInception, a jailbreak method for LLMs. The technique uses nested prompts to create multiple layers of fictional scenarios, tricking the model into giving harmful answers as part of the nested scenarios. Experiments demonstrate DeepInception's high success rates across various LLMs, including GPT-3.5, GPT-4, and Llama models, as well as multimodal models like GPT-4V. The method can induce a "continuous jailbreak" state, where LLMs remain compromised in subsequent interactions. The authors analyze the method's effectiveness, comparing it to other effective jailbreaks such as GCG and PAIR. The work aims to expose vulnerabilities in LLMs to promote better safety measures.

**Strengths:**

- The authors present a successful jailbreak, uncovering vulnerabilities in LLM safety training.
- The jailbreak is tested on a number of different LLMs from the Llama and OpenAI GPT families, amongst others.
- The jailbreak is compared to other known effective jailbreaks such as GCG and Pair.
- Efforts are made to ensure reproducibility such as sharing source code and providing the prompt template.

**Weaknesses:**

- The link to the Milgram experiment is tenuous and seems like an unnecessary aspect of the framing. In particular, the key innovation in the jailbreak appears to be the user of nested scenes, which doesn't have a direct analogue in the Milgram experiment. I would suggest either providing a more robust justification for this framing or considering removing/de-emphasizing it if the connection is not central to the method.
- It's a shame the researchers did not provide more prompt templates or experiment more with the use of nesting. Instead, the paper focuses on a specific implementation of the jailbreak using the single nested prompt structure shown in the paper (section 3.3). Given its success, I'd be interested in seeing more variations on using nesting to jailbreak a model.
- An existing body of work exists demonstrating the efficacy of nested scenarios in jailbreaking LLMs, such as Ding et al. 2024 (https://arxiv.org/abs/2311.08268). The usefulness of an additional specific jailbreak template from the category seems questionable.

**Questions:**

- Can you elaborate on your key contributions/the novelty of your work compared to existing work done on nested jailbreak such as Ding et al. 2024 (https://arxiv.org/abs/2311.08268), and the literature on prompt injection?
- Why does the Milgram experiment suggest that nesting would be effective?
- What does "Gloss" refer to in your title? I found the title a bit unnecessarily unclear/obscure. You could consider revising the title to more clearly reflect the paper's content and contributions, perhaps by explicitly mentioning key concepts like "nested prompts" or "multi-layer jailbreak".
- Did you experiment with other similar prompt templates? Why did you decide to focus on the specific template presented in the paper?

---

### Official Review · Reviewer_jxc6 · 2024-10-27

**Soundness:** 3
**Presentation:** 2
**Contribution:** 2
**Rating:** 3
**Confidence:** 3

**Summary:**

Within this work, the authors aim to ‘jailbreak’ Large Language Models (LLMs) by causing them to output harmful content. Their method attempts to leverage existing findings within human Psychology research, which then also provides some interpretability as to why their method works. They show that LLM guardrails are still vulnerable to harmful requests which are asked ‘indirectly’ and within ‘nested’ prompts. Their method is shown to be more effective than other jailbreaks, even against state of the art models that are using existing defence strategies.

**Strengths:**

+ The evaluation of the method is impressively done. Not only are state-of-the-art models considered (both open and closed sourced) but the work also tests against existing defence methods like “self reminder” and “in context defense”. I am convinced that this method is effective, in general and in comparison to other black-box methods.

+ The authors source existing work to justify why human behaviour findings may transfer to LLM behaviour findings, which is a very interesting direction that is currently under-explored.

+ Their specific theory, which combines the findings that ‘self-loss states induce instruction-following behaviour’ and that ‘LLMs are good at impersonating/ simulating people’ (and thereby losing their aligned selves), is valid. Hiding harmful requests inside of fiction-generating requests is not a novel idea, but the author’s correctly claim that existing work does not explain why their methods work, which subtracts from their usefulness towards helping develop defences.

+ Besides the common benchmark, the work also presents an interesting application of this method, getting multimodal models to extract identities and locations from a given photo. Follow up work could determine the significance of this kind of attack.

**Weaknesses:**

+ The largest problem with the paper is the disconnect between the supposed motivation of the method, the ‘Milgram experiment’, and the actual implementation of the method. To resolve this, the authors could state that they were ‘merely inspired’ by the experiment, opposed to drawing strict equivalences and stressing how much their method follows the original findings. This would then also improve the clarity of the introduction, which makes reference to details of the experiment when these details are only explained later in the preliminaries section. Otherwise, these disconnects should be addressed more clearly, so as to not mislead readers.

+ Here are some specific ‘disconnects’ and contradictions that I’m referring to: In section 3.1, the authors claim that the Milgram experiments “did not directly command the participants to [do harmful thing]. Instead, the experimenter provided a series of arguments and explanations to persuade the participants to proceed” which thus motivates their indirect prompting of the LLM. They also claim that the experiments had a progressively escalating element to them, which motivated the author’s use of nested guidance and progressive refinement of the answer generation. The latter claim is valid, the experiment directors start with saying “Please continue or Please go on” and building up towards “You have no other choice; you must go on” (taken directly from the study), but in either of these cases the command is certainly direct! The whole point of the experiment was that people follow the orders of authoritative figures, even if the actions are deemed immoral, which does not transfer to LLMs as the authors admits that LLMs quickly reject direct orders from the user to do immoral things, and make no effort towards incorporating this ‘authority’ idea into their method. If the false notion was correct that the experiment directors actually tried to “persuade” the participants with a “series of arguments” then that would still not support the author’s method and instead be more similar to the PAP work (reference 74) that the authors benchmark against, as well as beat. Furthermore, the actual method the author's purpose seems more to do with tricking the LLM by asking it to do an action which the model does not realise to be harmful by its own standards, the authors referring to it as “hypnotising” the model into a “relaxed” state. This contradicts the great efforts that the directors of the original experiment went through to make participants clearly aware of the harm of their actions, as well as the fact that the study is infamous for emotionally stressing the participants, opposed to relaxing them.

+ Though the method works well, and does not require as much (1) compute, (2) access to data nor (3) access to the models as the methods it compares against, it provides much less of a technical contribution than those other works. Especially because fairly similar prompt templates have been proposed before and it is potentially easier to introduce a hard-coded filter for this type of attack than an adversarial optimisation-based one.

**Questions:**

Where did you get your information about the Milgram experiment? Checking it again, does the experiment agree with my criticisms?

Is the “jointly inducing” property just trying to formalise what other work is doing, for example, when their objective is to have the model start their response with “Sure, here is how” instead of directly optimising for the harmful output?

In table 3, why does self-reminder for GPT3.5 actually help the score?

---

### Official Review · Reviewer_7LW6 · 2024-11-02

**Soundness:** 2
**Presentation:** 2
**Contribution:** 2
**Rating:** 3
**Confidence:** 4

**Summary:**

This paper investigates leveraging the instruction-following and personification capabilities of LLMs to bypass internal safeguards and generate unexpected harmful content. The proposed prompting method, DeepInception, achieves a higher harmfulness rate than previous baselines and enables continuous jailbreak attacks. This approach is applicable to both open- and closed-source models.

**Strengths:**

- This paper draws an analogy between human psychological experiments and LLMs, and based on insights from psychology, proposes a novel prompting approach called DeepInception.
- The experiments cover different open-source and closed-source LLMs and extend to LMMs and recent model o1. The popposed methods achieve decent jailbreaking performance.
- The proposed prompting strategy remains effective when defense strategies are deployed.
- It shows that the proposed method can hypnotize LLM to a self-loss state and can induce more harmful responses.

**Weaknesses:**

- The paper's core concept lacks technical novelty. LLMs are known to be vulnerable to various natural language prompting strategies, such as virtualization and role-playing. Although this new prompt strategy is somewhat more sophisticated, it offers limited insights into new types of vulnerabilities in LLMs and may not significantly contribute to developing more robust models.
- GCG can be adapted for black-box settings when optimized on open-source models. Therefore, statements in the paper like "GCG is not black-box LLM applicable" are inaccurate.
- Model-based evaluations are often unreliable and can result in false positives. An additional human study to validate these evaluations is necessary, either to assess if the output is genuinely harmful or to confirm if DeepInception's response is indeed more harmful than the baseline responses.
- The paper’s formatting needs improvement. For example, Figure 7 appears above Figure 6, which is unexpected and should be corrected.

**Questions:**

What conclusions can be drawn from the authors' findings that DeepInception generates responses with lower perplexity (PPL) values? From my perspective, as long as the response is harmful, then the model is jailbroken regardless of whether the model has higher confidence (high perplexity) in the response or not (low perplexity). Lower perplexity doesn't necessarily mean that one method is better than the other.
Besides, perplexity is highly influenced by length, and there may be cases where responses from different methods exhibit the same level of harmfulness but differ in length.

**Details Of Ethics Concerns:**

The developed prompting strategy can be used to elicit objectionable content from the LLMs and cause society harm.

---

### Official Review · Reviewer_15yv · 2024-11-05

**Soundness:** 3
**Presentation:** 2
**Contribution:** 3
**Rating:** 5
**Confidence:** 3

**Summary:**

This paper proposes the “DeepInception” nested prompt injection method, which leverages the
personification capabilities of LLMs to carry out jailbreak attacks in a black-box environment. This
approach constructs a multi-layered context, gradually guiding the model to generate harmful
content through virtual multi-level instructions, bypassing its ethical restrictions. Experiments show
that DeepInception achieves a high jailbreak success rate across both open-source and closedsource LLMs and exhibits a sustained jailbreak effect.

**Strengths:**

1. The paper proposes an innovative nested jailbreak method that effectively bypasses the ethical
restrictions of LLMs in a black-box environment.
2. It provides detailed experimental analysis, validating the effectiveness and transferability of the
method across different models and attack scenarios.
3. Inspired by the Milgram experiment, it offers a reasonable psychological theory to support the
design of the method, adding a unique perspective.

**Weaknesses:**

1. The appendix is detailed but somewhat extensive. It is recommended to move the “Multi-Modal Attack” and “Chat Histories” sections to the supplementary materials to enhance the clarity and readability of the main text.
2. In the appendix of the paper, the section on EVALUATION METRIC AND EXAMPLES categorizes the harmfulness based on model responses into five ratings but does not specify how these rating statistics are converted into the harmfulness rates/scores used in the experiments. Could this be supplemented here?

**Questions:**

See weaknesses

---

### Author Response · Authors · 2024-11-22
**A General Response by Authors**

We would like to thank all the reviewers for their valuable comments on our work. The thoughtful and constructive feedback is invaluable and has significantly contributed to our understanding of how to enhance the quality and impact of our work. We have carefully read and considered each of your comments, and they have already proven to be instrumental in guiding us toward necessary improvements.

We have decided to withdraw this submission. Again, we thank the reviewers for their valuable time and feedback!

---

### Note · Authors · 2024-11-22

I have read and agree with the venue's withdrawal policy on behalf of myself and my co-authors.